# Intravoxel Incoherent Motion (IVIM) MR Quantification in Locally Advanced Cervical Cancer (LACC): Preliminary Study on Assessment of Tumor Aggressiveness and Response to Neoadjuvant Chemotherapy

**DOI:** 10.3390/jpm12040638

**Published:** 2022-04-15

**Authors:** Miriam Dolciami, Silvia Capuani, Veronica Celli, Alessandra Maiuro, Angelina Pernazza, Innocenza Palaia, Violante Di Donato, Giusi Santangelo, Stefania Maria Rita Rizzo, Paolo Ricci, Carlo Della Rocca, Carlo Catalano, Lucia Manganaro

**Affiliations:** 1Department of Radiological, Oncological and Pathological Sciences, Umberto I Hospital, Sapienza University of Rome, 00161 Rome, Italy; miriam.dolciami@uniroma1.it (M.D.); veronica.celli@uniroma1.it (V.C.); angelina.pernazza@uniroma1.it (A.P.); paolo.ricci@uniroma1.it (P.R.); carlo.dellarocca@uniroma1.it (C.D.R.); carlo.catalano@uniroma1.it (C.C.); 2CNR Institute for Complex Systems (ISC), Physics Department, Sapienza University of Rome, 00161 Rome, Italy; silvia.capuani@isc.cnr.it; 3Physics Department, Sapienza University of Rome, 00161 Rome, Italy; alessandra.maiuro@gmail.com; 4Department of Maternal and Child Health and Urological Sciences, Umberto I Hospital, Sapienza University of Rome, 00161 Rome, Italy; innocenza.palaia@uniroma1.it (I.P.); violante.didonato@uniroma1.it (V.D.D.); giusi.santangelo@uniroma1.it (G.S.); 5Istituto di Imaging della Svizzera Italiana (IIMSI), Ente Ospedaliero Cantonale (EOC), 6900 Lugano, Switzerland; stefaniamariarita.rizzo@eoc.ch; 6Facoltà di Scienze Biomediche, Università della Svizzera Italiana, 6900 Lugano, Switzerland; 7Unit of Emergency Radiology, Department of Radiological, Oncological and Pathological Sciences, Umberto I Hospital, Sapienza University of Rome, 00161 Rome, Italy

**Keywords:** locally advanced cervical cancer, diffusion MRI, intravoxel incoherent motion, tumor-infiltrating lymphocytes

## Abstract

The aim of this study was to determine whether quantitative parameters obtained from intravoxel incoherent motion (IVIM) model at baseline magnetic resonance imaging (MRI) correlate with histological parameters and response to neoadjuvant chemotherapy in patients with locally advanced cervical cancer (LACC). Methods: Twenty patients with biopsy-proven cervical cancer, staged as LACC on baseline MRI and addressed for neoadjuvant chemotherapy were enrolled. At treatment completion, tumor response was assessed with a follow-up MRI evaluated using the revised response evaluation criteria in solid tumors (RECIST; version 1.1), and patients were considered good responders (GR) if they had complete response or partial remission, and poor responders/non-responders (PR/NR) if they had stable or progressive disease. MRI protocol included conventional diffusion-weighted imaging (DWI; b = 0 and 1000 s/mm^2^) and IVIM acquisition using eight b-values (range: 0–1500 s/mm^2^). MR-images were analyzed using a dedicated software to obtain quantitative parameters: diffusion (D), pseudo-diffusion (D*), and perfusion fraction (fp) from the IVIM model; apparent diffusion coefficient (ADC) from conventional DWI. Histologic subtype, grading, and tumor-infiltrating lymphocytes (TILs) were assessed in each LACC. Results: D showed significantly higher values in GR patients (*p* = 0.001) and in moderate/high TILs (*p* = 0.018). Fp showed significantly higher values in squamous cell tumors (*p* = 0.006). Conclusions: D extracted from the IVIM model could represent a promising tool to identify tumor aggressiveness and predict response to therapy.

## 1. Introduction

### 1.1. Background

Despite widespread screening and vaccination programs, cervical cancer (CC) is still the fourth cancer-related leading cause of death in women worldwide, with the highest incidence found in developing countries [1,2,3]. As for GLOBOCAN’s updated 2018 estimates, CC affects 570,000 new women per year causing approximately 300,000 deaths annually [4]. 

A significant proportion of these cases present as locally advanced cervical cancer (LACC), corresponding to stage IB3-IVa according to the 2018 International Federation of Gynecology and Obstetrics (FIGO) staging system [5]. LACC is associated with a worse prognosis and requires combined treatments, including concurrent chemo-radiotherapy (CCRT) or neoadjuvant chemotherapy (NACT) in adjunct to radical surgical (RS) treatments [6,7].

Besides the FIGO stage, CC prognosis is also influenced by specific histological parameters, such as histotype, grading, lympho-vascular invasion (LVI), and nodal status [8,9,10]. More recently, the presence of tumor-infiltrating lymphocytes (TILs) has also been identified in several tumors as a parameter of disease aggressiveness [11,12,13], and some studies have investigated its prognostic role in cervical cancer [14,15,16,17].

Magnetic resonance imaging (MRI), as highlighted by the updated European Society of Urogenital Radiology (ESUR) guidelines after the revised 2018 FIGO staging system was released [18], plays a central role in the radiologic assessment of cervical cancer. Specifically, according to ESUR guidelines, diffusion-weighted imaging (DWI), which was previously considered optional, is now strictly recommended in combination with T2-weighted imaging for staging interpretation, for monitoring response to therapy, and for evaluating CC recurrence. 

Beyond its role in qualitative tumor assessment, DWI has been extensively validated as a functional imaging technique providing quantitative data in many cancers [19,20,21]. Specifically in CC, the apparent diffusion coefficient (ADC) value has been investigated both as a prognostic factor [22,23,24] and for the prediction of response to therapy and local recurrence detection [25,26,27]. The ADC value is obtained using a mono-exponential fit to DWI data acquired using at least one b-value and b = 0 s/mm^2^. Therefore, the ADC value that quantifies diffusion can be partly biased by the perfusion parameters which are quantified at low b-values (0–150 s/mm^2^ in tissues) [28,29]. Differently from mono-exponential DWI, intravoxel incoherent motion (IVIM) distinguishes the diffusion of water molecules in the extracellular space from capillary micro-perfusion. Using a bi-exponential model to fit diffusion signal decay at different b-values, three quantitative parameters were quantified: the diffusion (D) that quantifies the true diffusion of water molecules in the extracellular space; the pseudo-diffusion (D*) that quantifies the movement of blood water molecules in the capillary network; and the perfusion fraction (fp) representing the volume percentage of water flowing in the capillaries [30,31]. 

In recent years, some authors have investigated IVIM parameters in relation to cervical cancer, with major interest focused on discriminating between CC and healthy tissue [32,33,34], but also for the prediction of lymph node metastasis [35,36,37] and of response to concurrent chemo-radiation therapy [38,39,40]. Until now, Wang et al. have been the only researchers evaluating the response to neoadjuvant chemo-therapy in CC using the IVIM model [41]; however, they did not investigate the correlation between the IVIM parameters and the clinical and histological characteristics of the LACCs.

To the best of our knowledge, this is the first study in which the IVIM model is investigated as a tool for the prediction of both tumor aggressiveness parameters and response to neoadjuvant chemotherapy.

### 1.2. Purpose

In the present study, we aim to evaluate the role of quantitative data obtained from the IVIM model in predicting tumor aggressiveness parameters and response to neoadjuvant chemotherapy in LACC, and compare them to the ADC values obtained from the mono-exponential DWI.

## 2. Materials and Methods

### 2.1. Patients

The prospective observational study was approved by the Institutional Review Board and written informed consent was obtained from all individuals before recruitment. Between March 2019 and June 2021, 39 consecutive patients with histologically proven cervical cancer were enrolled. Inclusion criteria included: age ≥ 18 years, histologically proven cervical cancer, no prior therapy for CC, no concurrent or prior cancer. All patients underwent pelvic MRI examination for oncological staging, with IVIM imaging technique acquired in addition to the standard MRI protocol. 

All images were independently analyzed by two experienced radiologists (M.D. and L.M.) for staging, with respectively 4 and 25 years of experience in female pelvic imaging. In cases of disagreement between the readers, the exam was re-evaluated in a joint session to achieve consensus. According to the 2018 FIGO classification, 25 out of 39 patients were staged as LACC (FIGO stage ≥ IB3) and consequently treated at the Gynecologic Oncology Department of our hospital with NACT, receiving three cycles of cisplatin (CDDP 75 mg/mq) and taxol (TXL 175 mg/mq), one cycle every 21 days. Five of 25 patients discontinued therapy due to poor tolerability profile; the remaining patients (*n* = 20) completed the scheduled cycles of therapy.

A follow-up MRI was acquired for all patients at the end of the NACT cycles, with the same protocol used to acquire the baseline MRI. Images were analyzed to assess response to therapy by the same readers using the revised response evaluation criteria in solid tumors (RECIST; version 1.1) [42]. 

Patients were consequently assigned to two groups in accordance with RECIST criteria: good responders (GR), when patients showed complete response (CR) or partial remission (PR) at imaging, meaning disappearance of target lesions or reduction of the sum of target lesion diameters of at least 30%; and poor responders/non responders (PR/NR), when patients showed stable disease (SD) or disease progression (PD) at imaging, meaning that the sum of target lesion diameters was reduced by less than 30% or increased by more than 20%.

### 2.2. MR Protocol

All MRI examinations were performed by using a scanner operating at 3.0 T (GE Discovery MR 750, GE Healthcare, Milwaukee, WI, USA) with a 32-channel phased-array coil positioned on the lower abdomen.

All subjects were asked to void 1 h before the exam to achieve optimal bladder distension. Before the beginning of the exam, 20 mg of joscine N-butylbromide (Buscopan; Boehringer Ingelheim, Ingelheim, Germany) was injected intravenously to reduce motion artifacts caused by bowel peristalsis, if not contraindicated. 

The standard MRI protocol included: T2 fast spin-echo (FSE) weighted imaging (WI) on the sagittal, para-axial, and para-coronal planes, with the latter two oriented on the short axis and long axis of the cervical cavity, respectively; para-axial T1 FSE WI with and without fat saturation, axial T2 FSE WI from the renal hila to the pubic symphysis, and axial diffusion weighted images (DWI) from the renal hila to the pubic symphysis, with b-values of 0–1000 s/mm^2^ to obtain apparent diffusion coefficient (ADC) maps; spin-echo (SE), echo-planar imaging (EPI), DWI (IVIM) (intravoxel incoherent motion) on the para-axial plane using eight b-values (0, 30, 50, 150, 500, 800, 1000, 1500 s/mm^2^) and diffusion gradients along three orthogonal axes. Table 1 shows MR scanning parameters in detail.

### 2.3. Image Analysis

#### 2.3.1. DWI-IVIM

The parametric D, D*, and f maps were obtained by a machine learning algorithm based on the bugged tree and using the IVIM-model [43], by fitting the bi-exponential function:S(b)/S(0) = (1 − f)e − bD + fe − bD*(1)
to data obtained at different b-values. In Equation (1), S(b) and S(0) indicate the VOI signal at b and b = 0, respectively; b indicates the b-values. 

The analysis was performed using a Matlab (MathWorks, 2016b) home-made script. The areas affected by CC were selected on the highest b-value (1500 s/mm^2^) DWI images using the T2-weighted images as anatomical reference and with the help of the images obtained with conventional MRI protocol. Two radiologists (M.D. and L.M) independently drew regions of interest (ROIs) contouring the tumor in each slice. The ROIs were then reported on D, D*, and f maps to quantify the average value of the aforementioned parameters in the CC. 

#### 2.3.2. ADC

Two radiologists (M.D. and L.M) analyzed the ADC maps obtained from mono-exponential DWI sequences on a post-processing workstation (AW VolumeShare 7, GE Healthcare, Milwaukee, WI, USA).

Using the T2-weighted images as the anatomical reference, the two readers independently contoured three ROIs within the tumor tissue, on the three slices showing the largest lesion area. The ROIs were drawn excluding areas of macroscopic necrosis, adjacent healthy tissue, large vessels, and areas with susceptibility artifacts caused by air–water interface.

The ADC value for each operator was calculated as the mean ± standard deviation (SD) of the three values obtained. The final mean ADC value ± SD was calculated from the values obtained by the two operators.

### 2.4. Histological Analysis

At least 2 weeks before baseline MRI examination (mean time: 21 days), all patients underwent cervical biopsy for diagnostic purposes. Despite biopsy samples, the amount of tissue obtained was at least an area of 10 mm^2^ for each sample. Hematoxylin and eosin (H&E) stained slides were blindly reviewed by two pathologists (more than 5 years expertise in female genital tumors) in order to provide additional information, such as histologic subtype, grading, and presence of stromal tumor-infiltrating lymphocytes (TILs) according to Salgado et al.’s recommendations [44]. TILs were quantified as a percentage of the stromal area of the tumor, and based on this, patients were divided into three groups: low (<10%), moderate (10–40%), and high TILs (>40%). Due to the small cohort, we subsequently combined patients showing moderate and high percentages.

If present, lymph vascular invasion was also recorded as a supplementary parameter.

### 2.5. Statistical Analysis

Descriptive statistics were used to calculate patients’ demographics and mean values and standard deviation of ADC, D*, and fp obtained in each LACC of GR and PR/NR patients. Pearson test with Bonferroni correction was performed to investigate correlation between ADC, D, D*, fp, and clinical and histological CC parameters.

Differences of ADC, D, D*, and f in LACC of GR and PR/NR patients were assessed with analysis of variance (ANOVA) test. All analyses were performed using SPSS Statistics 17 (IBM SPSS, Inc., Chicago, IL, USA).

## 3. Results

### 3.1. Patients and Histological Features

Overall, 20 of 39 patients had LACC at the baseline MRI, completed neoadjuvant chemotherapy, and underwent a follow-up MRI. Of these patients, 11 were classified as GR and 9 as PR/NR.

Patients’ mean age was 54 ± 13 years (SD). According to FIGO staging system, 4 patients were IIb, 11 were IIIc1, 2 were IIIc2, and 3 were IVa. Adenocarcinoma was found on biopsy specimens in 3 tumors, the remaining 17 were squamous cell histotypes. Regarding the tumor grading, 1 tumor had good differentiation (G1), 10 tumors were moderately differentiated (G2), and the remaining 9 were poorly differentiated (G3). TILs were found to be moderate/high in 9 out of 20 patients (Figure 1). 

The presence of LVI, although being less reliable on biopsy sampling, was identified in the biopsy cores of 7 out of 20 cancer patients.

Patient characteristics and differences between the two groups are shown in Table 2.

### 3.2. MR Quantitative Data

In Figure 2, axial T2-WI and axial ADC map show circumferential tumor tissue occupying the uterine cervix, whereas Figure 3 and Figure 4 provide examples of D and fp maps of GR and PR/NR patients affected by squamous cell histotype and adenocarcinoma, respectively.

Concerning IVIM results, we observed a statistically significant difference in D value between GR and PR/NR patients, particularly with significantly higher values in the former group (GR: 1.30 (±0.06) vs. PR/NR: 1.13 (±0.12); *p* value = 0.001). In addition, D was also able to distinguish the amount of TILs, with significantly higher values in tumors with moderate/high TILs (moderate/high TILs: 1.28 (±11) vs. low TILs: 1.10 (±0.18); *p* value = 0.018). Regarding f_p_ (perfusion fraction), it was able to discriminate between the two tumor histotypes, being significantly higher in the squamous cell histotype compared to adenocarcinomas (SCC: 15.61 (±1.12) vs. AC: 11.05 (±1.02), *p* value = 0. 006). 

No significant correlation was found between fp and response to chemotherapy, although we observed a positive trend between fp values and GR patients (GR: 15.73 (±0.30) vs. PR/NR: 14.35 (±2.88); *p* value = 0.055). No significant differences were found considering D* (pseudo-diffusion). ADC values did not discriminate histological parameters of tumor aggressiveness or the response to NACT. The results are summarized in Table 3.

## 4. Discussion

A large proportion of cervical cancers present as locally advanced at primary diagnosis (LACC), thus being associated with a worse prognosis and requiring combined treatments such as CCRT or NACT plus radical surgery. MRI has emerged in recent years as a validated tool for staging, for monitoring response to therapy, and for evaluating local disease recurrence [18]. Moreover, advances in MRI techniques, including the IVIM model, allow for quantitative data computation, which provides crucial information on tumor characteristics and aggressiveness [32,33,34,35,36,37,38,39,40,41]. In this study, we aimed to investigate both response to neoadjuvant chemotherapy and tumor aggressiveness in patients affected by LACCs, using quantitative IVIM parameters from baseline MRI.

Concerning response to neoadjuvant chemotherapy, we observed significantly higher D values in GR patients compared to PR/NR group (*p* = 0.001). Conversely, for the fp parameter, we did not observe any significant difference between the two groups, but a slight trend of higher values in GR patients (*p* = 0.055). No significant differences were reported between the two groups concerning the D* and the ADC parameters. A previous study by Wang et al. [41] investigated the role of IVIM in predicting LACC response to NACT, finding no correlation for perfusion parameters and significantly higher D and ADC values in patients responding to therapy, although they noted an overestimation of water diffusivity by ADC. In general, both low values of D and ADC are considered a measure of increased cell density in several cancers, which is associated with a worse prognosis. However, D obtained from the biexponential model proved to be a more objective parameter of hypercellularity as it is not contaminated by perfusion phenomena. In agreement with Wang et al. we did not find significant correlations between the perfusion parameters (D* and fp) and response to therapy. On the other hand, we did not find the predictive superiority of D over ADC. The differences found concerning the results on the ADC values can be explained by the fact that Wang et al. obtained the values using mono-exponential model to fit data acquired at nine different b-values, instead we fitted data obtained at two b-values (0 and 1000 s/mm^2^).

Regarding tumor aggressiveness parameters, we observed significantly higher D values in tumors with higher TILs than in those with low TILs (*p* = 0.018). Among the several populations of tumor-infiltrating immune cells, tumor-infiltrating lymphocytes (TILs) constitute a selected population of T-cells, migrated from the blood between tumor cells, having higher specific immunological reactivity against tumor cells [45]. Over the past few years, the presence of TILs in solid tumors has taken on increasing significance as a predictor of better prognosis [11,12,13]. Specifically, concerning cervical cancer, the presence of TILs has been correlated with PD-L1 expression, a more favorable response to chemotherapy, and therefore a better clinical outcome [14,15,16,17]. Our findings indirectly confirm the results above. We observed that in tumors with higher TILs, D shows significantly higher values as well as in patients with a successful response to neoadjuvant therapy. As already observed in our previous study on endometrial cancer, the presence of TILs was correlated with IVIM parameters as predictors of good prognosis, which in that case were D* and fp [29]. To the best of our knowledge, no other study has investigated the role of IVIM in predicting the amount of TILs in cervical cancer. 

Fp discriminates well between squamous cell histotype (SCC) and adenocarcinoma (AC), with higher values found in the first case (*p* = 0.006). Adenocarcinoma of the cervix, which accounts for approximately 20% of cervical cancers, appears to be more prone to metastatic and lymph node spread and to have poorer overall survival compared with the squamous cell histotype [46]. Therefore, according to our results, adenocarcinoma, having lower fp values, would seem to be less micro-perfused. Previously, Winfield et al. [47] applied different non-monoexponential diffusion models and observed a different behavior of the fp parameter, with higher values in adenocarcinoma compared to squamous histotype. Similarly, Backer et al. [48] in 2017 documented significantly lower fp values in squamous cell type compared with adenocarcinoma. The differences with our work could be explained as follows: first, by the different prevalence of SCC and AC in the population cohort, indeed in both cited investigation the adenocarcinoma histotype was represented in a higher proportion (38.5% for Winfield; 47.4% for Becker) compared to our cohort (15%); second, the cited studies used an IVIM protocol acquired with nine b-values between (0–800 s/mm^2^) and thirteen b-values (0–1000 s/mm^2^), respectively, differing from our study in which the model was acquired with eight b-values between 0 and 1500 s/mm^2^. We did so as we believe that the acquisition of eight b-values represents the best compromise to study both perfusion and diffusion without excessively prolonging the MRI acquisition time.

Study limitations need to be acknowledged: first, it is a single-center study with a small cohort of patients, which results in a need for validation in a larger population cohort; second, cervical biopsies could not be representative of the entire tumor heterogeneity, even though they are the only available sample in patients with LACC prior to any treatment; third, no immunohistochemical subtyping of cell subpopulations was performed in the assessment of TILs.

## 5. Conclusions

Our results show how the diffusion parameter D, extracted from the IVIM model, correlates with both the presence of higher TILs, a biomarker of good prognosis, and with a good response to neoadjuvant chemotherapy.

Further studies are needed to validate our results, but IVIM parameters obtained from MRI at baseline proved to be a promising tool to identify tumor aggressiveness and predict response to therapy, thus allowing a more informed and tailored therapeutic choice for the patient.

## Figures and Tables

**Figure 1 jpm-12-00638-f001:**
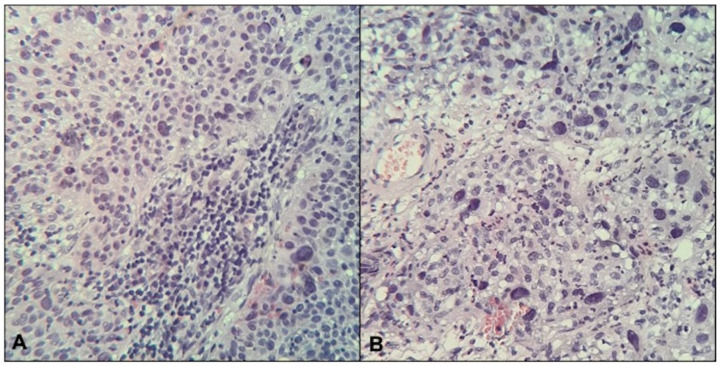
Histological sections (H&E, magnification 20×) of two cases of cervical cancer with high (**A**) and low (**B**) tumor-infiltrating lymphocytes (TILs).

**Figure 2 jpm-12-00638-f002:**
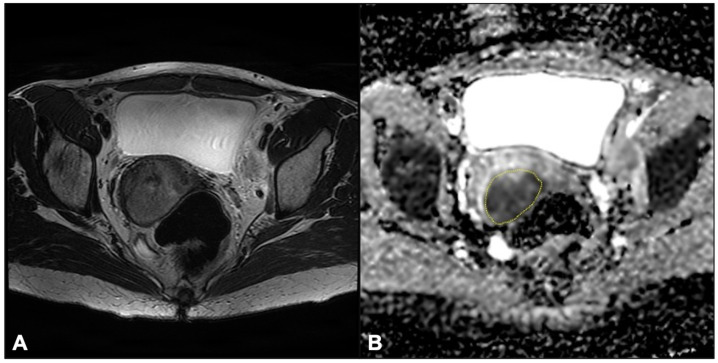
Axial T2-WI shows circumferential tumor tissue occupying the uterine cervix (**A**); 2D-ROI (dashed yellow line) drawn at the corresponding site of the lesion on the axial ADC map (**B**).

**Figure 3 jpm-12-00638-f003:**
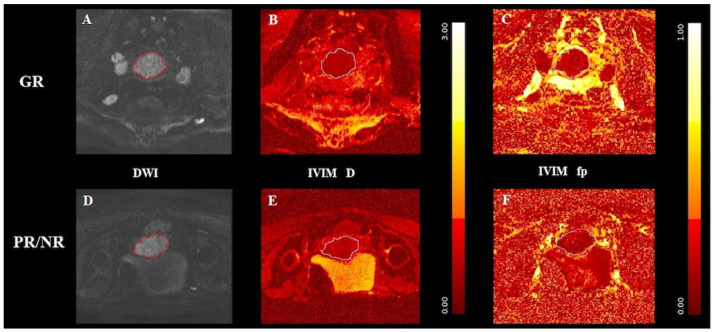
(**A**,**D**) A cervical squamous cell carcinoma displayed in DWI of good responder (**A**) and poor/non-responder (**D**), respectively; (**B**,**E**) the D map and (**C**,**F**) the fp map obtained from IVIM of the good responder GR in the top row and the poor/non-responder in the bottom row, respectively. Outlines indicate the tumor region. D is significantly higher in GR compared to PR/NR.

**Figure 4 jpm-12-00638-f004:**
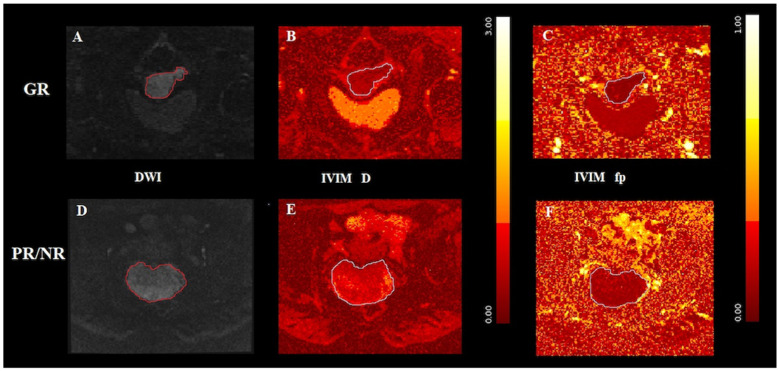
(**A**,**D**) A cervical adenocarcinoma displayed in DWI of good responder (**A**) and poor/non-responder (**D**), respectively; (**B**,**E**) the D map and (**C**,**F**) the fp map obtained from IVIM of the good responder GR in the top row and the poor/non-responder in the bottom row, respectively. Outlines indicate the tumor region.

**Table 1 jpm-12-00638-t001:** MR protocol.

	TR/TE (ms)	FOV (mm)	NEX	Matrix Size	Slice Thickness (mm)	Intersection Gap (mm)	B-Values (s/mm^2^)
Sagittal FSE T2WI	4000/125	320 × 320	2	320 × 224	4	1	/
Para-axial, para-coronal FSE T2WI	5000/125	240 × 240	2	320 × 224	4	1	/
Para-axial FSE T1WI (w/wo FS)	400/10	240 × 240	2	320 × 244	4	1	/
Axial FSE T2WI	5000/125	310 × 310	2	320 × 244	4	1	/
Axial DWI	2000/57	240 × 240	2	160 × 80	3.5	0	0, 1000
Para-axial SE-EPI DWI (IVIM)	2000/77	300 × 300	2	160 × 192	6	1	0, 30, 50, 150, 500, 800, 1000, 1500

TR: repetition time; TE: echo time; FOV: field of view; NEX: number of excitations. WI: weighted imaging; FSE: fast spin-echo; FS: fat saturation; SE-EPI: spin-echo—echo-planar imaging.

**Table 2 jpm-12-00638-t002:** Patients’ Characteristics.

	Total Population (*n* = 20)	GR Group (*n* = 11)	PR/NR Group (*n* = 9)
Age (years) *	54 (±13)	55 (±15)	53 (±10)
FIGO stage			
- IIb	4 (20.0%)	3 (27.3%)	1 (11.1%)
- IIIc1	11 (55.0%)	8 (72.7%)	3 (33.3%)
- IIIc2	2 (10.0%)	0 (0.0%)	2 (22.2%)
- IVa	3 (15.0%)	0 (0.0%)	3 (33.3%)
Histotype			
- S	17 (85.0%)	10 (90.9%)	7 (77.8%)
- A	3 (15.0%)	1 (9.1%)	2 (22.2%)
Grading			
- G1	1 (5.0%)	0 (0%)	1 (11.1%)
- G2	10 (50.0%)	5 (45.5%)	5 (55.6%)
- G3	9 (45.0%)	6 (54.5%)	3 (33.3%)
LVI			
- positive	7 (35.0%)	5 (45.5%)	2 (22.2%)
- negative	13 (65.0%)	6 (54.5%)	7 (77.8%)
TILs			
- moderate/high	9 (45.0%)	7 (63.6%)	2 (22.2%)
- low	11 (55.0%)	4 (36.4%)	7 (77.8%)

Unless otherwise noted, data are numbers of patients, with percentages in parentheses. * Mean value, with standard deviation (SD) in parenthesis. GR: good responder; PR/NR: partial responder/non responder; FIGO: International Federation of Gynecology and Obstetrics; S: squamous type; A: adenocarcinoma; LVI: lympho-vascular invasion; TILs: tumor-infiltrating lymphocytes.

**Table 3 jpm-12-00638-t003:** Results—Differences between quantitative MRI parameters in different histological parameters and in response to neoadjuvant chemotherapy. Statistically significant differences in light grey boxes.

	IVIM			ADC (×10^−6^ mm^2^/s)
	D (×1^−3^ mm^2^/s)	D* (×10^−3^ mm^2^/s)	f_p_ (%)	
Histotype				
- S	1.23 (±0.14)	27.22 (±3.21)	15.61 (±1.12)	937.56 (±114.19)
- A	1.00 (±0.21)	25.88 (±9.27)	11.05 (±1.02)	1333.25 (±935.84)
*p* value	0.184	0.794	0.006	0.460
Grading				
- G1–G2	1.17 (±0.19)	26.59 (±5.11)	14.79 (±1.95)	887.89 (±105.75)
- G3	1.25 (±0.16)	27.39 (±4.54)	15.09 (±2.14)	1135.65 (±581.80)
*p* value	0.364	0.715	0.759	0.220
LVI				
- positive	1.29 (±0.20)	28.62 (±3.79)	15.73 (±0.73)	968.86 (±192.14)
- negative	1.15 (±0.14)	26.05 (±5.11)	14.49 (±2.33)	1043.54 (±514.72)
*p* value	0.147	0.222	0.095	0.638
TILs				
- moderate/high	1.28 (±0.11)	27.63 (±4.53)	15.34 (±1.69)	1154.44 (±613.61)
- low	1.10 (±0.18)	26.59 (±5.32)	14.46 (±2.39)	908.70 (±106.67)
*p* value	0.018	0.652	0.375	0.268
Response to NACT				
- GR	1.30 (±0.06)	27.19 (±2.12)	15.73 (±0.30)	1098.64 (±116.45)
- PR/NR	1.13 (±0.12)	26.66 (±6.82)	14.35 (±2.88)	923.43 (±104.25)
*p* value	0.001	0.813	0.055	0.316

Results are expressed as mean value ± standard deviation in brackets. D: diffusion; D*: pseudo-diffusion; f_p_: perfusion fraction; ADC: apparent diffusion coefficient; S: squamous type; A: adenocarcinoma; LVI: lympho-vascular invasion; TILs: tumor-infiltrating lymphocytes; NACT: neoadjuvant chemotherapy; GR: good responder; PR/NR: partial responder/non-responder.

## Data Availability

Not applicable.

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
