# Peer review of "Intravoxel Incoherent Motion (IVIM) MR Quantification in Locally Advanced Cervical Cancer (LACC): Preliminary Study on Assessment of Tumor Aggressiveness and Response to Neoadjuvant Chemotherapy"

_jpm, 2022, doi:10.3390/jpm12040638_

Round 1

Reviewer 1 Report

While the subject matter of using intravoxel incoherent motion was a concept introduced around 1986, this manuscript entitled “Intravoxel Incoherent Motion (IVIM) MR quantification in Locally Advanced Cervical Cancer (LACC): assessment of tumor aggressiveness and response to neoadjuvant chemotherapy” truly captures its utility in using a number of determined IVIM/baseline MRI quantitative parameters and of their possible correlation with both histology and neoadjuvant chemotherapy responses in patients with LACC. Incorporating experienced independent radiologists for staging, follow-up MRIs, and assignment of patients as being “good responders” or “poor responders/non responders”, is both well-described. Evaluated data related to the quantitative parameters of diffusion, pseudo-diffusion, perfusion fraction from the model, and apparent diffusion coefficient from conventional diffusion-weighted imaging was well-presented and supported with MRI images, analysis of apparent diffusion coefficient maps, and histology.  Special note is made of Figure 4 which compared maps of a good and poor/non-responder, respectively.

Minor suggestion, though possibly missed by this reviewer, is to include “…diffusion-weighted imaging (DWI, b=0 and 1000s/mm2)” in Abstract, if possible, for some readers not so familiar with the DWI abbreviation. 

Author Response

We would like to thank the reviewer for the valuable review work, changes have been incorporated into the text accordingly to his suggestion.

Reviewer 2 Report

This manuscript is a fascinating study evaluating chemo-sensitivity for locally advanced cervical cancer by Intravoxel Incoherent Motion (IVIM) MR quantification using MRI.

1. First, we recommend examining tumor response according to RESIST criteria: a reduction of 50% or more is considered a substantial response to chemotherapy, which is usually rarely seen. Why don't evaluate using the tumor markers, such as serum SCC, CA125, and CEA.

2. Second, please reconsider how to evaluate the presence or absence of Tumor-Infiltrating Lymphocytes. It is not sufficient to confirm the presence or absence of Tumor-Infiltrating Lymphocytes in a single biopsy specimen, as shown in Figure 1. Please evaluate it using a scoring system for TILs in line with the prior literature, and please validate the part about evaluating TILs, as this is the crux of this paper.

Author Response

1. We would like to thank the reviewer for the valuable comment, the changes have been included both in the abstract and in the main text, adding also a reference for the RECIST criteria.

As for tumor markers, these were not homogeneously available for all patients, so we decided not to include them in subsequent analyses.

2. We would like to thank the reviewer for the valuable comment.

Regarding TILs scoring system, we followed the recommendations by Salgado et al. and quantified TILs as a percentage of tumor stromal area: low (<10%), moderate (10-40%), and high TILs (>40%).

Because our patient cohort was relatively small, we combined patients with high and moderate TILs. Following your precious advice, we made it explicit in the text accordingly and also added a reference.

We agree with the reviewer that the TIL's evaluation is the crucial result of our study. This was a pilot study, meant to present the feasibility of the method, and we are currently validating these preliminary findings in a different larger-cohort project. Therefore, the additional findings will be presented as the ongoing project is finished.

  1. English revision has been performed.

Round 2

Reviewer 2 Report

Dear Authors,

If the study is preliminary, please modify the title as "A preliminary study". 

Author Response

We thank the Reviewer for the suggestion, we have revised the title congruently.
